# Chromatographic Determination of Total Selenium in Biofortified *Allium* *sp.* following Piazselenol Formation and Micro-Solid-Phase Extraction

**DOI:** 10.3390/molecules26216730

**Published:** 2021-11-06

**Authors:** Bogdan M. Bosca, Augustin C. Mot

**Affiliations:** Chemometrics and Bioanalytical Chemistry Laboratory, Analytica Research Center, Department of Chemistry, Faculty of Chemistry and Chemical Engineering, Babeș-Bolyai University, 11 Arany János Street, RO-400028 Cluj-Napoca, Romania; bogdan.bosca@stud.ubbcluj.ro

**Keywords:** selenium, *Allium*, piazselenol, micro-solid-phase extraction

## Abstract

Herein, a method based on selective piazselenol formation is applied for total selenium determination in biofortified *Allium* species. Piazselenol is formed by reacting Se(IV) with an aromatic diamine, namely 4-nitro-1,2-phenylenediamine, in acidic medium. Samples were digested in a nitric acid/hydrogen peroxide open system, followed by selenate reduction in hydrochloric acid. Reaction conditions were optimized in terms of pH, temperature, reaction time, and other auxiliary reagents for interference removal, namely, EDTA and hydroxylamine. For the extraction of the selectively formed 4-nitro-piazselenol, micro-solid-phase extraction (μSPE) was applied, and the analysis and detection of the corresponding complex was performed by HPLC coupled with DAD. An external standard calibration curve was developed (R^2^ = 0.9994) with good sensitivity, and was used to calculate the total selenium content from several *Allium* plants material, with good intermediate precision (RSD% < 16%). The accuracy of the method was evaluated using both, a comparison with an accepted reference method from our previously published data, as well as three certified reference material with recoveries between 84–126%. The limit of detection was determined to be 0.35 μg/g (in solids) and 1.1 μg/L (in solution), while the limit of quantification was 1.07 μg/g and 3.4 μg/L (in solution). Using the proposed method, selenium content can be quickly and accurately determined in several types of samples. In addition, this study present experimental conditions for overcoming the interferences that might be encountered in selenium determination using piazselenol.

## 1. Introduction 

Selenium (Se) was discovered in 1817 by Jöns Jacob Berzelius, and is a metalloid placed in group 16, period 4 of the Periodic Table [1,2,3]. In terms of chemical properties, selenium is similar to sulfur and tellurium [2,3]. Selenium is an essential micronutrient in metazoan, naturally distributed in all compartments of the environment—soil, water, air, plants—and also in humans and animals [4,5,6]. Compared to other micronutrients, there is a narrow range between the recommended intake (40 µg/day) of selenium and the dose at which this element becomes a toxicant for humans (400 µg/day) [7]. Toxicity of selenium is strongly linked to the chemical form of the element [2,3]. Inorganic species of selenium, selenite (SeO_3_^2−^) or selenate (SeO_4_^2−^), are characterized by up to 40 times higher toxicity than organic compounds of selenium such as selenomethionine (SeME) and selenocysteine (SeCys) [8]. A concentration of selenium that is below of the recommended intake can produce serious disorders [1]—Keshan disease [9], liver necrosis, muscular dystrophy [10], reproductive disorders [11]—but an excess of selenium results in poisoning [9,12].

Regarding to its physiological activity, selenium is well known for its antioxidant properties [3,13] and in living organisms throughout enzymes as glutathione peroxidase [2,11], thioredoxin reductase, and deiodinases [3,14]. It is also involved in selenoprotein synthesis and thyroid metabolism [15].

Deficiency of selenium is a serious problem in some geographical areas. A solution for this issue is the biofortification of plants with selenium, in the form of selenate, through selenium soil enrichment. In order to identify the areas where biofortification should be mandatory, several analytical techniques for detection of selenium needed to be proposed, optimized, and validated [3,16]. Its essentiality for animals, but also its toxicity and the narrow range between the normal intake and the concentration that makes selenium a toxicant, are the main reasons why a selective and sensitive analytical method for the detection of selenium from various sample types is demanded. The present method represents a good candidate for this kind of research, characterized by satisfying values of limit of detection (0.35 μg/g in solid and 1.1 μg/L in solution), recovery (84–126%), and RSD <16%, characteristics which will be discussed in further detail in the following sections.

There are numerous analytical methods used for detection of selenium with remarkable performances (e.g., gas chromatography (GC) [5], the fluorimetric method based on piazselenol formation [4], voltametric methods [17], induced coupled plasma mass spectrometry (ICP-MS) [6], and several types of atomic absorption spectrometry (AAS) [18]).

Compared to the validated techniques generally used for the detection of selenium, the proposed method does not require special conditions (e.g., clean room for ICP-MS), or specially trained people to use the necessary equipment in order to be applied. Moreover, the required analysis can be carried out more quickly, and for smaller volumes of solution.

The preparation step of the sample plays a crucial role in further steps of the analytical technique performed. Some generally used preparation methods are represented by liquid-liquid extraction (LLE) [5], liquid phase microextraction (LPME) [19], single-drop microextraction (SDME) [20], and dispersive liquid-liquid microextraction (DLLME) [21]. In this study, the preparation step of the samples is represented by μSPE, performed using C18 cartridges.

The main purpose of this work is the optimization and validation of an analytical method for the determination of selenium in biofortified *Allium sp.* following piazselenol formation and solid face microextraction (μSPE).

## 2. Materials and Methods

### 2.1. Chemicals

Double distilled water was used for obtaining solutions in this work. Nitric acid (65%) from Merck, (Darmstadt, Germany), hydrogen peroxide from Chempur (Piekary Śląskie, Poland), hydrochloric acid and acetonitrile from Honeywell Fulka (Seelze, Germany), ammonia (25%) from VWR Chemicals, (Fontenay-sous-Bois, France), ethylenediaminetetraacetic acid (EDTA) in the form of a disodium salt dihydrate from Raral (Budapest, Hungary), and 4-nitro-1,2-phenylenediamine from Sigma Aldrich (St. Louis, MO, USA) were used. Hydroxylamine hydrochloride, sodium selenate (anhydrous, >99.8%, metal basis), and sodium selenite (anhydrous, >99.75%, metal basis) were obtained from Alfa Aesar (Kandel, Germany), and dimethyl sulfoxide was purchased from VWR International (Solon, OH, USA).

### 2.2. Apparatus, Analytical Instrumentation and Chromatographic Procedures

Lyophilization was conducted using a CHRIST, ALPHA 1–2 LD plus lyophilizer. For grinding the lyophilized *Allium* samples, a Retsch MM400 ball mill was used. The mineralization process was performed using a standard sand bath. Samples’ thermal treatment was carried out using HM100-Pro thermoblock. Micro-solid-phase extraction (μSPE) technique was conducted, using a cartridge RIDA C18. Moreover, a XH-D Vortex Mixer was utilized in order to obtain homogenous mixtures, and for centrifugation, Zentrax 280 E Centurion Scientific K3 Series was used. Chromatographic separations were performed on an Agilent Technologies 1100 HPLC Series system (Agilent, Santa Clara, CA, USA) equipped with a G1322A degasser, a G13311A binary gradient pump, a column thermostat, a G1313A autosampler, and a G1316A UV detector. For the separation, a reverse phase column was used (Zorbax SB-C18, 100 × 3.0 mm i.d., 3.5 μm particle from Agilent Technologies (Santa Clara, CA, USA)). A mixture of ultra-pure water acidified with 0.1% trifluoroacetic acid (solvent A) and acetonitrile (solvent B) was used as a mobile phase, under gradient elution. Two gradient programs were used. The first short gradient program that was used for synthetic simple samples was composed of a starting isocratic period, 20% B from 0 to 2 min, followed by a gradient from 20% to 100% B, from 2 to 8 min for another isocratic period of 100% B from 8 to 9 min, and a sharp return from 100% to 20% B from 9 to 9.1 min, followed by an isocratic equilibration of the column with 20% B from 9.1 to 10 min. The second gradient program was derived from first program after several optimizations that were required for real samples that are much more complex. This optimized second gradient program consisted of a much longer isocratic initial step at 22% B from 0 to 12 min, a gradient from 22–90% B from 12 to 12.1 min, an isocratic step at 90% B from 12.1 to 13 min, and a sharp gradient step from 90 to 22% B from 13 to 13.1 min, followed by an isocratic column equilibration at 22% B from 13.1 to 14 min. The flow rate was 1.5 mL/min, and the injection volume 10 µL for all the samples and both chromatographic procedures described above. The chromatogram was monitored at 344 nm, while the DAD detector was set to register full spectra in the 210–510 nm range every 2 s. The column temperature was set to 30 °C.

### 2.3. Optimization of the Working Conditions

Before total selenium analysis from the real samples was conducted, and in order to establish the optimum conditions for the piazselenol formation from selenite, some synthetic samples were tested under different conditions regarding pH, time, and incubation temperature. Influence of other auxiliary reagents such as EDTA and hydroxylamine—which were required for interference reduction—were tested on real samples. The reaction parameters for piazselenol formation were optimized for obtaining the best analytical conditions in terms of recovery and precision, as well as in the shortest time possible. First, the pH dependence was tested, with the concentration of hydrochloric acid varying from 0.3 mM to 4.0 M. The other parameters were kept constant: 1 μg Se (as selenite) in 2 mL vials, 0.3 mM of 4-nitro-1,2-phenylenediamine, 30 min incubation time at 90 °C. Moreover, the reaction was conducted in several acidic conditions, with the concentration of nitric acid varying between 0.5 M and 6 M, and the concentration of hydrochloric acid maintained at a constant of 1 M. Regarding the temperature effect on piazselenol formation, several measurements were conducted at two values of temperature near room temperature (20 °C and 30 °C), and two values at elevated temperature (70 °C and 90 °C). The other reaction parameters were kept constant: 1 μg Se (as selenite) in 2 mL vials, 0.3 mM of 4-nitro-1,2-phenylenediamine, HCl 0.1 M, and 30 min incubation time. In the case of real samples—that were obtained through concentrated nitric acid digestion as described in next section—prior to 4-nitro-1,2-phenylenediamine addition, a concentrated ammonia solution (25%) was used for pH adjustment to the optimum value. In order to find the appropriate concentration of ammonia that needed to be added, its concentration was varied between 0 and 6 M. The other reaction parameters were kept constant: 90 min incubation time at 30 °C. Since EDTA are required to be added for the reduction of interference of several cations, the optimum ratio between ammonia and EDTA was assessed. In the case of EDTA, the values of concentration were 5, 10, and 15 mM, and for every concentration level of EDTA, the ammonia concentration was varied at 0, 3, and 6 M. Due to overlapping peaks of coelutes with the piazselenol, in the case of complex real samples, hydroxylamine hydrochloride addition was tested at 0.6 M in order to reduce this interference effect.

### 2.4. Analytical Procedure for Real Samples Analysis

The validated and optimized method that was developed in this work was applied on *Allium* species such as *Allium fistulosum*, *Allium ampeloprasum*, *Allium ursinum*, *Allium schoenoprasum*, *Allium cepa*, and *Allium triquetrum*. *Allium sp.* plants were grown in a phytochamber in controlled light (150 μmol/m^2^/s), temperature (22 °C), and humidity (70%), using a 12 h day/night cycle. The plants were biofortified with selenium in the form of selenate solution at three levels of concentration (1, 5, and 20 mg Se/L) through selenium soil enrichment. The mature plants were lyophilized and subsequently milled. The green powder obtained was weighted to an appropriate amount in Berzelius beakers (typically 120 mg), and 2 mL of concentrated nitric acid (65%) was added. This mixture was kept for 24–48 h at room temperature. Following this, 1 mL of concentrated hydrogen peroxide (30%) was added. Sample digestion was carried out in open system, by introducing the beakers containing the samples in a sand bath for about 2 h at cca. 150–200 °C. The beakers containing the samples were constantly covered with watch glasses. After digestion, the cooled samples were quantitatively transferred into a 10 mL volumetric flask, and diluted to the mark with distilled water. The next step was a reduction of selenate to selenite, since only selenite reacts with the aromatic diamine. In order to perform the reduction, a volume of 600 µL of the sample (transferred from the 10 mL volumetric flask into a 2 mL sealed vial) was treated with 600 µL of concentrated hydrochloric acid (37%) in the closed, sealed 2 mL vial, and the resulted mixture was incubated 60 min at 96 °C in a thermoblock. After incubation, the obtained mixture was treated with 200 µL of hydroxylamine hydrochloride (5.5 M), in the same 2 mL vial. Prior to the diamine addition, 400 µL of concentrated ammonia (25%) was added (in the same 2 mL vial) in order to adjust the pH to the optimum value, and 120 µL of EDTA (250 mM) was added in order to avoid the effect of possible cations interferences. The final reagent added was 4-nitro-1,2-phenylenediamine (10 µL, 300 mM, dissolved in dimethyl sulfoxide) and, after that, the entire mixture was incubated for 90 min at 30 °C. After every addition of a reagent, the respective mixture finding in 2 mL closed vials was carefully vortexed.

After incubation, the samples were centrifuged, and μSPE was performed using a C18 cartridge. Before usage, the cartridges were treated with acetonitrile and equilibrated with 500 µL of a solution of HCl 0.1 M, and, after that the entire sample (2 mL) was passed through the cartridge, followed by acidified water washing. In order to eliminate water, the cartridge was centrifuged (1500 rpm, 3 min). After the water was eliminated from the cartridge, elution was performed using acetonitrile (two times with 100 µL each time). To collect the eluted liquid, the cartridges were centrifuged (1500 rpm) after every addition of acetonitrile. Liquid phase eluted was transferred into HPLC vials, and was analyzed using the HPLC technique. All the above-detailed steps are summarized in Figure 1.

### 2.5. Analytical Calibration, Figures of Merit, Method Validation and Statistical Analysis

For both chromatographic procedures described in Section 2.2, an external standard method was applied for the development of the calibration curve, in the domain of 0–13 ng injected Se—equivalent to 0–0.5 μg Se/mL—using standard solutions of sodium selenite. All of the conditions for piazselenol formation, as well as the sample preparation—SPE binding and elution—were identical with those used for the real samples. After the chromatographic analysis, the peak area of the piazselenol was measured using the Agilent ChemStation software, followed by the application of least squares method in Excel or Statistica 13 software in order to obtain the mathematical equation of the calibration curve. The limit of detection was calculated from the standard deviation of residuals of the calibration curve (s_y/x_), i.e., LOD = 3 × s_y/x_/m, where m is the slope of the calibration curve. Intermediate precision (within lab reproducibility) was evaluated over four months (*n* = 3), each with a separate calibration curve. Accuracy of the method was tested using both a reference method with (HG-HR-CS-QFAAS), a comparison of the results from the same samples with previously published data, as indicated in Table 1, and three certified reference materials.

## 3. Results and Discussion

Piazselenol formation from selenite and an aromatic diamine, in an acidic medium (Figure 2), is one of the most selective reactions for inorganic Se(IV) species, and has been long used in analytical chemistry for selenium determination from various environmental and biological samples [10,22]. Piazselenol is generally detected via GC-MS and, to a lesser extent, using HPLC. In this work, we have developed and optimized an analytical procedure for selenium determination mainly from plants—soil, multivitamin and mushroom samples have also been analyzed—using HPLC-DAD. Various parameters which influence the piazselenol formation were optimized, in order to find the best analytical conditions. The optimized parameters were the following: pH, temperature, time and the concentration of ammonia, and EDTA. 

In the present study, the pH was optimized, with the concentration of hydrochloric acid ranging between 3 mM and 4 M. As can be observed in Figure 3A, a hydrochloric acid concentration higher than 0.01 M facilitates the formation of the 5-nitropiazselenol complex, while lower values of concentration lead to poorer recovery level. Therefore, the optimum value of hydrochloric acid concentration that was selected for further experiments was 0.05 M, which corresponds to a pH equal to 1.3. Knowing the optimum value of the pH, piazselenol formation was performed in several acidic mixtures in order to find the appropriate ration and type of acids to be used, especially for real sample preparation. A mixture of acids which contained hydrochloric acid and nitric acid—typical reagent used in sample digestion—was tested, with a varying concentration of nitric acid, and a maintained constant concentration of hydrochloric acid (Figure 3B). The reason for this approach was to find out the nitric acid’s influence on the formation and stability of piazselenol, since nitric acid is used in the digestion of plant-based samples. The graph shows that the formation of 5-nitropiazselenol was not significantly affected by the acidic mixture that contained a lower concentration of nitric acid (0.5 M). However, at an elevated concentration of nitric acid (higher than 1 M), new chromatographic peaks appeared, and piazselenol formation was drastically affected. On the other hand, if only one of the two acids is used—nitric acid or hydrochloric acid—the reaction works well (Figure 3A,B).

Piazselenol formation takes place within a wide range of temperatures, providing sufficient reaction time. As can be observed in Figure 4A,B, the reaction was conducted at four temperature values (20 °C, 30 °C, 70 °C, and 90 °C) for 30 min, and 5-nitropiazselenol was formed in all these cases, the best recovery being obtained at a temperature of 90 °C, during a 30 min reaction duration. However, due to the presence of side compounds and interferences in real samples, which can decompose or take part in secondary reactions (at a high temperature), a temperature of 30 °C was selected, while increasing the incubation time to 90 min was selected for further experiments. The following analyzed certified reference materials, as well as other plant based real samples, reveal that the selected reaction time and temperature were optimum (see below).

As indicated in the Materials and Methods section, the analyzed real samples were digested in an open system using concentrated nitric acid and hydrogen peroxide, and the formed selenate was reduced to selenite—since only this form leads to piazselenol—using concentrated hydrochloric acid. Therefore, in following these steps, the samples were highly acidic. In order to adjust the pH of the reaction mixture to near the optimum value, ammonia was added, before an addition of 4-nitro-1,2-phenylenediamine. Plant-based samples contain numerous other interferences, such as iron, copper, and other cations, which can also react with 4-nitro-1,2-phenylenediamine, and produce some side products. In order to avoid the effect of these interferences, EDTA was added. Moreover, the ratio between the concentrations of these two reagents influences the reaction and the recovery level (Figure 5A). As can be observed in Figure 5, the magnitude of the non-piazselenol peaks strongly depends upon the ammonia and EDTA concentrations. While the first peak (near 2 min elution time) depends rather on the acidity of the solution, the second dominant peak (near 4.5 min elution time) is significantly diminished at higher concentration of EDTA, especially after reducing the acidity with ammonia. In the case of samples much more complex than the plant-based ones, such as soil and multivitamin drugs, which contain much higher amounts of metal, some interferences peaks coeluted with the piazselenol, thus affecting the accuracy of the method. In this case, addition of hydroxylamine hydrochloride eliminated those peaks, and also reduced the other non-piazselenol peaks, as can be observed in Figure 5B. Hydroxylamine is a good reducing agent for several cations, and it was previously used to reduce bromine from other analytical methods involving piazselenol formation [23]. Nevertheless, there are some non-piazselenol peaks in chromatogram from Figure 5, which represent a proof of the high complexity of the analyzed samples.

Using the optimum values of reaction parameters (pH = 1.3, T = 30 °C, t = 90 min), a calibration curve was developed. The measurements were performed, with the concentration of Se (IV) ranging from 0.005 to 0.5 mg/L, and a linear increase in the analytical signal can be observed in Figure 6.

For the detection of selenium in the form of selenite from real samples, optimized parameters (pH = 1.3, T = 30 °C, t = 90 min, 0.6 M hydroxylamine, 15 mM EDTA, and 1.5 mM 4-nitro-1,2-phenylenediamine) were used. The method was applied for six species of *Allium* biofortified with different levels of selenium in the form of selenate via soil enrichment. Representative chromatograms for *Allium ampeloprasum* are shown in Figure 7. The peak corresponding to 5-nitropiazselenol can be observed in the case of *Allium ampeloprasum* for all levels of biofortification—the stronger the biofortification level, the higher the chromatographic peak, as expected—while no detection could be found in the case of non-biofortified sample. The total selenium content in all the analyzed *Allium sp.* samples is presented in Table 1, alongside the results obtained with HG-HR-CS-QFAAS, as a reference method, as previously described [24].

A very good agreement is observed between the piazselenol-based method as described in this study and the reference HG-HR-CS-QFAAS method, as can be observed in Table 1. In the case of a non-biofortified *A. ampeloprasum* sample containing very low Se (0.3 μg/g), for the piazselenol method as presented in this work, such low Se content is below the limit of detection. The last five *Allium* species were analyzed only through the piazselenol method. Furthermore, the intermediate precision of the method was assessed, and the results expressed as RSD% indicate an excellent precision of the method (Table 1). Higher RSD% values are observed for *A. ursinum* and *A. triquetrum,* which have distinctly longer and more fibrous stems, leading to a powder which is less homogeneous as compared to the other *Allium sp.*

It can also be observed that there is a difference, in terms of selenium uptaking, between the studied *Allium sp.* The main forms of selenium which are found in soils are selenite and selenate, selenate being the form in which selenium is assimilated by plants in most cases, and the chemical species that was used for soil enrichment in this study. Selenate enters the plants through their roots using sulfate transporters paths. These transporters are membrane proteins encoded by several genes. After its root assimilation, selenate is reduced to selenite, and subsequently, the selenite is reduced to selenide. Selenide is assimilated first as selenocysteine, and as selenomethionine after that. These selenoaminoacids can form proteins which can be toxic for the plant or can be methylated, methylation being a characteristic of selenium uptake. The methylation occurs directly from selenocysteine (to dimethyl selenide) in the case of accumulating species, and from selenomethionine (to dimethyl selenide) in the case of non-accumulating species. These two different mechanisms are a possible explanation for differences in the amount of selenium assimilated by the *Allium sp.* in our experiments. Regarding the selenium sequestration, it is strongly correlated with the rate of volatilization of this element. If the rate is high, the selenium will be released into the air, decreasing its concentration in the corresponding plant.

In order to further test the accuracy of the method, three different certified reference materials (CRMs) were analyzed, using the same procedure as described for the analysis of the *Allium* species (CSM-3Mushroom Powder, SRM 3280 Multivitamin, CRM 025050 Metals in soil) (See Appendix A). The chromatograms of the three analyzed CRMs are presented in Figure 7 B. Well defined and separated peaks of 5-nitropiazselenol are visible, while the other chromatographic peaks depend on the complexity of the CRMs (Appendix A). Table 2 presents the total selenium content in CRMs, as obtained using the described piazselenol-based method. As can be observed, in terms of accuracy, these results are in accordance with certified values.

Compared to other literature data (Table 3), the proposed method provides a higher value of the limit of detection, especially for spectrometric methods. Despite this, the optimized and validated method in this paper has the advantage of working with much lower volumes of sample in a shorter time.

## 4. Conclusions

The main purpose of this research was to develop and optimize a fast, simple, and sensitive method for the determination of selenium in vegetal samples. This analytical method is based on the reaction between selenite and an aromatic diamine in order to form a piazselenol complex. μSPE was applied on the formed piazselenol, and HPLC analyses were performed. The method was successfully conducted for several biofortified species of *Allium*, and was validated by measuring the content of selenium in three CRMs. It can be concluded that this method, based on piazselenol formation following μSPE leads to a highly efficient and rapid method for detection of selenium in vegetal samples characterized by LOD = 0.35 μg/g (in solid) and LOD = 1.1 μg/L (in solution), and a recovery of between 84 and 126%.

## Figures and Tables

**Figure 1 molecules-26-06730-f001:**
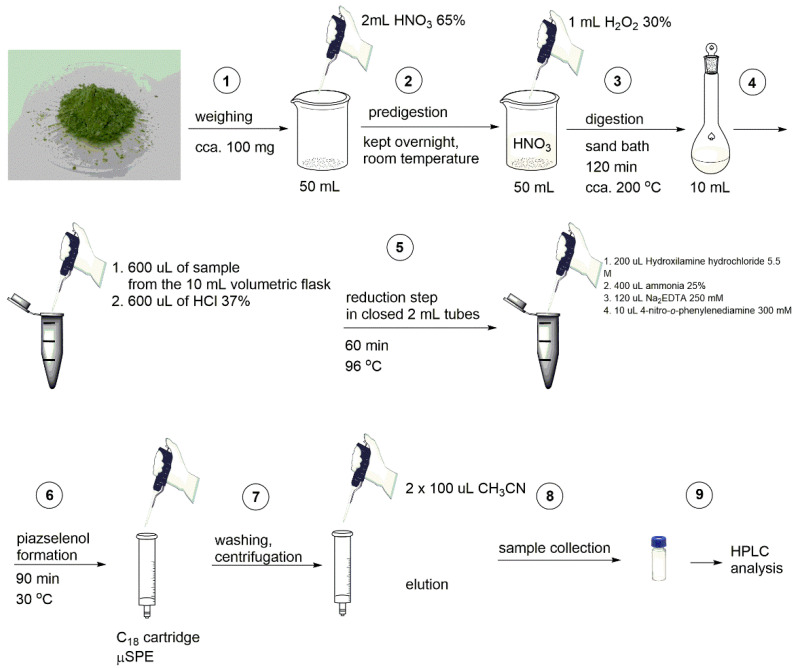
Schematic representation of the principal steps of the proposed method.

**Figure 2 molecules-26-06730-f002:**
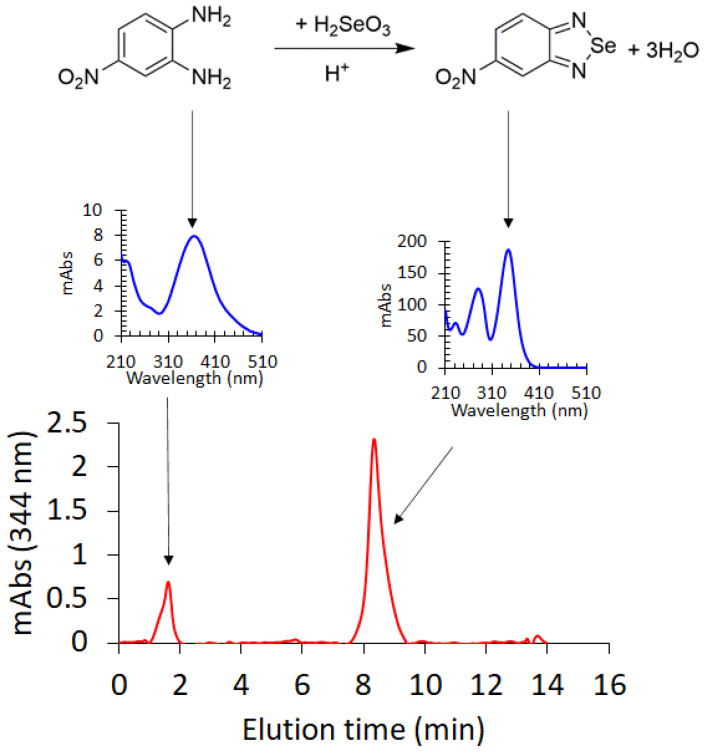
Reaction which describes 5-nitropiazselenol formation from 4-nitro-1,2-phenylenediamine and Se(IV), in acidic medium. Chromatogram that shows the peaks of 4-nitro-1,2-phenylenediamine excess eluted before minute 2, and 5-nitropiazselenol eluted just after minute 8. UV-Vis spectra (as measured by DAD) indicating the corresponding chromatographic peaks of 4-nitro-1,2-phenylenediamine excess (370 nm) and 5-nitropiazselenol (344 nm).

**Figure 3 molecules-26-06730-f003:**
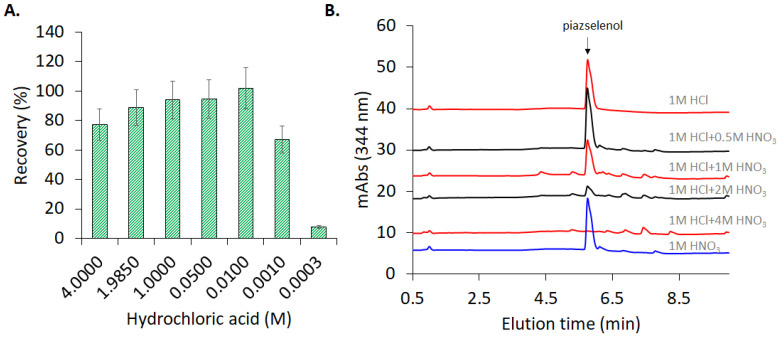
(**A**) Formation of 5-nitropiazselenol using different concentrations of hydrochloric acid (0.0003–4 M) at 90 °C for 30 min and keeping the concentrations of selenite (0.5 ppm) and 4-nitro-1,2-phenylenediamine (0.3 ppm) constant. (**B**) Formation of 5-nitropiazselenol in different acidic conditions.

**Figure 4 molecules-26-06730-f004:**
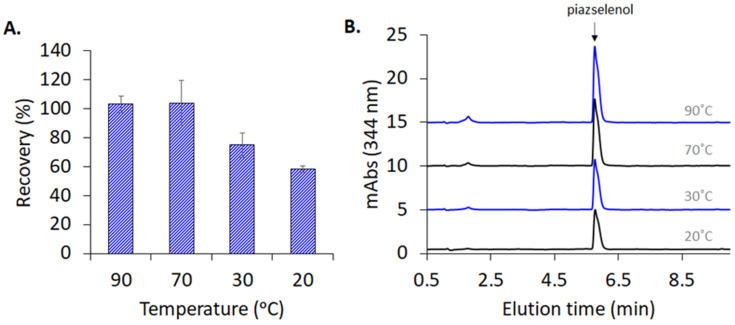
(**A**) Effect of the temperature of Se recovery, detected as 5-nitropiazselenol. Derivatizing conditions: 0.5 mg/L Se(IV), 0.3 mM of 4-nitro-1,2-phenylenediamine, 0.1 M of HCl for 30 min. (**B**) Chromatograms of 5-nitropiazselenol obtained in the indicated range of temperature (20–90 °C).

**Figure 5 molecules-26-06730-f005:**
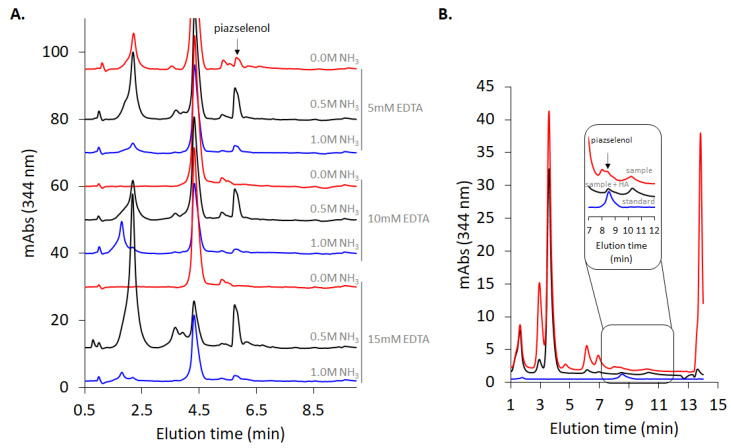
(**A**) Chromatograms presenting 5-nitropiazselenol peak at different ratio between ammonia and EDTA for the determination of Se in biofortified *Allium ampeloprasum*. (**B**) Influence of hydroxylamine addition (0.6 M) in the reaction mixture, in the case of SRM 3280 multivitamin certified reference material.

**Figure 6 molecules-26-06730-f006:**
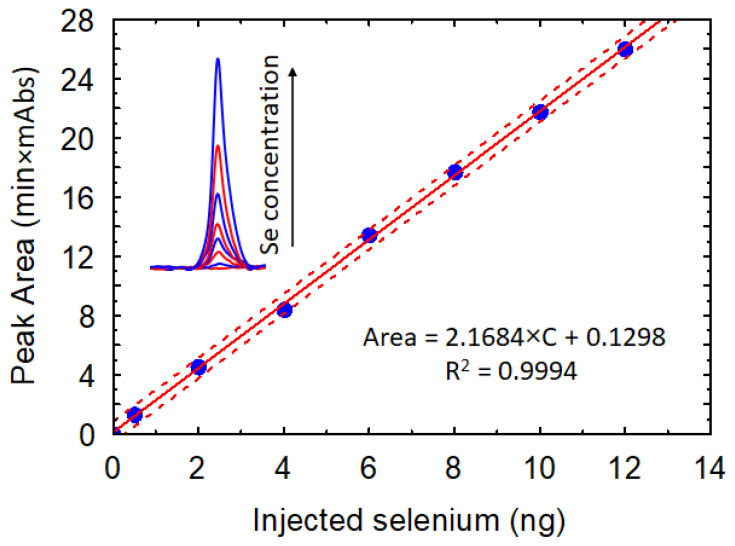
Calibration curve plotted using the following reaction parameters: 0.005, 0.02, 0.04, 0.06, 0.08, 0.1, 0.12, 0.25, and 0.5 mg/L of selenite. A volume of 2 mL of each standard solution was used for piazselenol formation (0.3 mM of 4-nitro-1,2-phenylenediamine, 0.05 M of HCl at 30 °C for 90 min), eluted in 0.2 mL acetonitrile from the μSPE cartridge, injected 10 μL.

**Figure 7 molecules-26-06730-f007:**
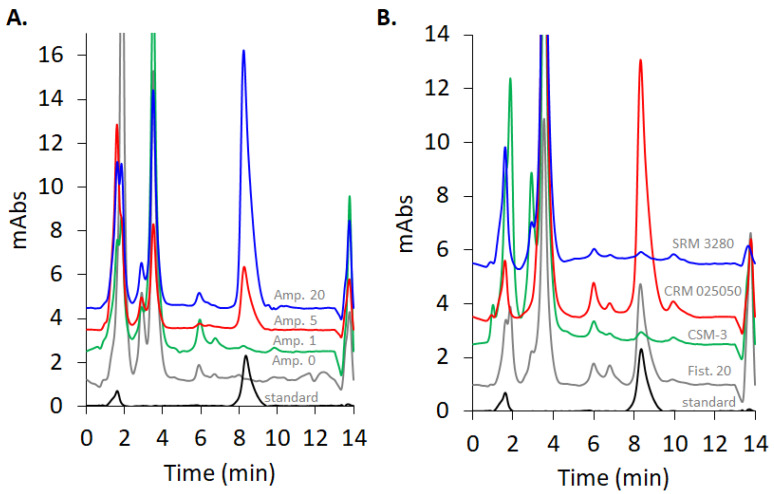
(**A**) Chromatograms of the *Allium ampeloprasum* (Amp.) samples at different level of biofortification according to the Table 1. Lower chromatogram is a Se(IV) standard sample with a clear 5-nitropiazselenol peak. (**B**) Chromatograms of the three certified reference materials and a biofortified *Allium fistulosum* (Fist. 20—according to Table 1) sample, alongside a Se(IV) standard.

**Table 1 molecules-26-06730-t001:** Total selenium content in the analyzed *Allium* samples with different levels of biofortification.

Sample	Level of Biofortification (ppm)	Concentration, Mean ± CI (µg/g) HG-HR-CS-QFAAS [24]	Concentration, Mean ± CI (µg/g) Piazselenol Method	RSD (%) Piazselenol Method
*A. fistulosum*	0	<LOD	<LOD	<LOD
*A. fistulosum*	1	28.5 ± 3.0	28.6 ± 3.4	9.7
*A. fistulosum*	5	270 ± 22	283.3 ± 23.1	8.2
*A. fistulosum*	20	1376 ± 79	1058.6 ± 49.3	4.6
*A. ampeloprasum*	0	0.30 ± 0.04	<LOD	−
*A. ampeloprasum*	1	43.5 ± 2.1	35.8 ± 3.1	7.1
*A. ampeloprasum*	5	429 ± 38	342.0 ± 8.8	2.1
*A. ampeloprasum*	20	1147 ± 84	1336.9 ± 95.2	5.8
*A. ursinum*	2.5	−	9.5 ± 2.1	15.8
*A. fistulosum*	2.5	−	15.0 ± 0.6	2.5
*A. schoenoprasum*	2.5	−	16.7 ± 0.9	3.0
*A. cepa*	2.5	−	18.1 ± 1.1	3.5
*A. triquetrum*	2.5	−	23.4 ± 4.2	12.7

**Table 2 molecules-26-06730-t002:** Total selenium content in soil, mushroom and multivitamin certified reference materials.

CRM	CSM-3MushRoom Powder	SRM 3280 Multivitamin	CRM 025050 Metals in Soil
Certified value ± U (µg/g)	17.43 ± 1.36	17.42 ± 0.45	518 ± 31
Found value ± CI (µg/g)	16.8 ± 3.1	17.1 ± 3.1	503.8 ± 45.7
Recovery ± CI (%)	95.3 ± 19.0	98.0 ± 17.8	98.0 ± 9.4
Composition	As, Cd, Cr, Cu, Hg, Pb, Se, Zn	B, Ca, Cl^−^, Cu, I, Fe, Mg, Mn, P, K, Zn, As, Cr, Pb, Se, Mo, Ni	Al, Sb, As, Ba, Be, B, Cd, Ca Cr, Co, Cu, Fe, Pb, Mg, Mn, Hg, Mo, Ni, K, Se, Si, Ag, Na, Sr, Tl, V, Zn

**Table 3 molecules-26-06730-t003:** Comparison of analytical parameters with other selenium determination methods from literature.

Sample	Method	Detector	Linear Range	LOD	LOQ	Recovery (%)	Reference
Food, dietary supplements, soil, water	HG-HR-CS-QFAAS	ContrAA 300 spectrometer	n.d.	0.062 µg/g	0.188 µg/g	77–99	[24]
Plasma, urine, water	Three phase HF-LPME-HPLC-UV	Varian 9050 UV-Vis detector	0.05–200 µg/L	0.02 µg/L	0.05 µg/L	95–103	[25]
Mushroom	RP-HPLC	UNICO WFZ UV-2100 spectrophotometer	0.12–12.0 μg/mL	0.06 μg/mL	n.d.	96.4–103.8	[26]
Water	Fluorimetric	Aminco-Bowman spectrofluorometer	0.02–1.0 µg	0.1 μg/L	n.d.	n.d.	[27]
Water	Liquid-liquid microextraction	Shimadzu GC 2010 + Electron-capture detector (GC–ECD)	0.015–10 μg/L	0.005 µg/L	n.d.	95–105	[21]
Vegetal samples	HPLC	G1316A UV detector	0.1–1.5 μg/mL	0.35 µg/g (1.1 μg/L)	1.07 µg/g (3.4 μg/L)	84–126	This paper

n.d.—not determined.

## Data Availability

The data might be shared upon request.

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
