# Peer review of "Chromatographic Determination of Total Selenium in Biofortified Allium sp. following Piazselenol Formation and Micro-Solid-Phase Extraction"

_molecules, 2021, doi:10.3390/molecules26216730_

Round 1

Reviewer 1 Report

In my opinion, it is a very interesting study and well-written manuscript which fits well to the scope of the Molecules Journal. I just have one little suggestion:

-write references f.ex. [1,2] at the end of the sentence and dot”.” behind it. You can also keep it in the middle of the sentence but not at the beginning.

Tables and graphs are very good and in my opinion, the manuscript can be published in the current form.

Reviewer 3 Report

The manuscript describes a novel method to quantify elemental selenium in plant tissue by acid digestion followed by chelation with an organic aromatic compound to yield a chromophore. The chromophore is then extracted and concentrated using standard reverse phase liquid chromatographic sample preparation and analytical techniques. Precision and accuracy of the novel method are presented, including appropriate analyses of Certified Reference Materials. The specifics of the methodology, and the optimisation process by which they were established, are described in appropriate detail. The figures are informative and clear. The manuscript is technically excellent and a significant contribution to the field.

I would like to see minor revisions regarding detail and specifics of the method. Also, the English language of the manuscript requires some modest amendment but is legible and mostly clear. I would like the following specifics addressed prior to acceptance of the manuscript:

  1. Please specify the brand, model, and dimensions of the LC column used.
  2. It is not clear from the text what happens between reduction in the 10 mL volumetric flask and addition of the final diamine reagent in 2 mL vials. 1 indicates that 600 µL of sample is transferred to the 2 mL vial at this point. Please clarify the written methodology here.
  3. Figure 1 would be improved by a continuous flow of arrows, or a delineation by numerals from step to step, to indicate the flow of the process.
  4. L235 – the Latin vide infra is delightful, but not a phrase I am familiar with. So, in the interests of simplicity, the English translation should be used instead.
  5. In Table 1, to which values does the RSD (%) column apply? I think that parameter should be calculated for both columns. Why is it so high? Typical targeted analytical methods aim for RSD <5% in intra-day and <15% in inter-day variation. Values in the 1,000s are a little alarming. Apologies if I am misinterpreting the data, but that is surely symptomatic of a need for clarification. Further apologies if I am mistaken, but I have a suspicion that you have calculated RSD for the combined results of the two methods, which would not be at all appropriate.
  6. Also Table 1: If you use <LOD for values in the Hg-HR-CS-QFAAS column then you should use the same notation for unquantifiable values in the piazselenol column.
  7. The discussion of the results in Table 3 claim that “Comparing to other literature data…this method has a lower limit of detection compared to other HPLC methods”. The LOD of the method presented here is 1.1 µg/L. LOD for the method in row 2 (reference 25) is 0.02 μg/L - substantially lower than your LOD. Please clarify this discrepancy.

I have the following comments not pertinent to acceptance:

  1. The manuscript would be improved by a brief discussion of the mechanisms of Se uptake and sequestration in the different species analysed. Consideration of the, apparently significant, differences in the observed uptake and sequestration of Se would be welcome.
